

# Harmonized soil database of Ecuador (HESD): data from 2009 to 2015

Daphne Armas[1,2], Mario Guevara[3], Fernando Bezares[1,4], Rodrigo Vargas[5], Pilar Durante[1,2,6], Víctor Osorio[7], Wilmer Jiménez[8], Cecilio Oyonarte[1,2]

[1]Departamento de Agronomía, Edif. CITEIIB. Universidad de Almería. 04120 La Cañada, Almería. España.

[2]Centro Andaluz para la Evaluación y Seguimiento del Cambio Global (CAESCG). Universidad de Almería.

[3]Centro de Geociencias - Universidad Nacional Autónoma de México. Campus Juriquilla, Qro. MX.

[4]Fundación CESEFOR, Soria España.

[5]Department of Plant and Soil Science, University of Delaware, Newark, DE, USA

[6]Agresta Sociedad Cooperativa. C/ Duque de Fernán Núñez, 2, 1º. 28012 Madrid. España.

[7]Escuela Superior Politécnica del Litoral. Facultad de ingeniería Marítima y Ciencias del Mar. Guayaquil.

Ecuador

[8]Ministerio de Agricultura y Ganadería, Dirección de Generación de Geoinformación Agropecuaria, Quito, Ecuador

*Correspondence to:* Cecilio Oyonarte (coyonart@ual.es); Daphne Armas (daphne.armas@gmail.com)

**Abstract.** One of the largest challenges with soil information around the world is how to harmonize archived soil data from different sources and how make it usable to extract knowledge. In Ecuador there have been two major projects that provided soil information, whose methodology, although comparable, did not coincide, especially regarding the structure of how information was reported. Here, we present a new soil database for Ecuador, comprising 13 542 soil profiles with over 51 713 measured soil horizons, including 92 different edaphic variables. Original data was in a non-editable format (i.e., PDF) making it difficult to access and process the information. Our study provides an integrated framework combining multiple data analytic tools for the automatic conversion of legacy soil information from analog format to usable digital soil mapping inputs across Ecuador. This framework allowed to incorporate quantitative information of a broad set of soil properties and retrieve qualitative information on soil morphological properties collected in the profile description phase, which is rarely included in soil databases. A new harmonized national database was generated using specific methodology to rescue relevant information. National representativeness of soil information has been enhanced compared to other international databases, and this new database contributes to filling the gaps of publicly available soil information across the country. The database is freely available to registered users at
https://doi.org/doi:10.6073/pasta/1560e803953c839e7aedef78ff7d3f6c (Armas, et al., 2022).



## 1 Introduction

There is an increasing need for updated soil datasets across the world. These datasets are required to develop soil monitoring baselines, soil protection, and sustainable land-use strategies and better understand soil response to global environmental change. Soil datasets are one of the most critical inputs for Earth system models (ESMs) to address different processes, such as the terrestrial carbon sinks and sources of greenhouse gases (Luo et al., 2016; Pfeiffer et al., 2020). Furthermore, access to spatially explicit, consistent, and reliable soil data is essential for digital soil mapping and evaluate the status of soil resources with an increased resolution to respond and assess global issues such as food security, climate change, carbon sequestration, greenhouse gas emissions, degradation through erosion and loss of organic matter or nutrients (FAO, 2015; FAO and ITPS. 2015 Pfeiffer et al., 2020). Unfortunately, one of the biggest challenges for digital soil mapping is the limited available information (e.g., soil profile descriptions, soil sample analysis) representing soil variability across the world.

In the last years, there have been growing efforts to improve the quality of soil datasets (Pfeiffer et al., 2020, Orgiazzi et al., 2018, Hengl et. al., 2017), specially we can find efforts to increase access to harmonized products containing comparable and consistent datasets. Global initiatives such as World Soil Information Service (WoSis, Batjes et al., 2020) or SoilGrids250m (Hengl et. al., 2017), for global pedometric mapping provide increasing soil information. Arrouays et al., 2017 affirm that over 800 thousand soil profiles have been rescued and collected into a database during the past decades, but only a small fraction (117 thousand) is accessible or shared with the international community. According to Batjes et al., 2019, large numbers of soil profiles stored in many country databases are yet not standardized and harmonized according to a global standard and are not shared; therefore, they are not available for use at a national level and less at a global level.

As acquiring new soil data is laborious and expensive, legacy databases and soil information historically collected are extremely valuable (Gray et al., 2015; Arrouays et al., 2017). This information is useful to test how soils have changed over time, but it usually comes from various projects that used different procedures, laboratory methods, standards, scales, taxonomic classification systems, and geo-referencing systems. Therefore, data must be rescued, compiled, and processed into a standard, consistent, and harmonized datasets which is a challenging process (Arrouays et al., 2018).

It is necessary to have consistent and spatially explicit information on different soil properties beyond the soil organic carbon (SOC) content, and reality shows the existence of a severe deficit of coherent information at regional, national, and global levels (Arrouays et al., 2017). Rossiter (2016) points out as primary deficits the scarce availability and the lack of harmonization that limits legacy database interoperability with global approaches. It is understood interoperability as the collective effort with the ultimate goal of sharing and using the information to produce knowledge and apply knowledge gained by removing conceptual, technological, organizational, and cultural barriers (Vargas et al., 2017). These efforts must come from various actors and institutions, including government ministries/agencies, the scientific community, landowners, civil society groups, and business owners.

It is vital to model the status of soil resources globally to an increasingly detailed resolution to have a better response and evaluate global and local issues, like soil salinization, land degradation and desertification (Pfeiffer et al.; 2020, FAO, 2015, Hengl et al., 2014). A harmonized soil information database will improve

the estimation of current and future land potential productivity, help identify land limitations, and identify land degradation risks, particularly soil erosion (Nur Syabeera et al. 2020). It also will contribute with scientific knowledge for planning a sustainable transformation of agricultural production and guiding policies to address emerging land competing issues concerning food production, bio-energy demand, and biodiversity threats (Montanella et al., 2016; FAO, 2015; McBratnet A., 2014). A harmonized soil

information database is of critical importance for rational natural resource management, making progress towards eradicating hunger and poverty, and addressing food security and sustainable agricultural development, especially concerning the threats of global climate change and the need for adaptation and mitigation (FAO/IIASA/ISRIC/ISS-CAS/JRC, 2009).

In Ecuador there have been two main efforts that have collected national soil information, one by the Instituto Espacial Ecuatoriano (IEE), and another by the Ministerio de Agricultura y Ganadería within the Sistema Nacional de Información de Tierras Rurales e Infraestructura Tecnológica (MAGAP-SIGTIERRAS) (Tracasa-Nipsa, 2015). These projects have comparable methodologies but there are substantial differences, especially on how the information is structured and presented. We have identified

over 13 500 soil profiles (and 51 713 measured soil horizons) that can be used to support digital soil mapping efforts across the country and the world (Loayza, et al. 2020). We highlight that so far this information has not been available to the scientific community and currently only 94 Ecuadorian soil profiles are included in global soil information services such as WoSis (Batjes et al., 2019).

The main objective of this study is to synthesize and harmonize available soil profile information collected between 2009 and 2015 across Ecuador. In this way, we develop a new soil database that is proposed to constitute a soil information system at the national scale following international standards for archiving and sharing soil data. Thus, this dataset can be easily integrated into global soil information systems. In addition, we provide an integrated framework combining various data analytic tools to convert legacy soil

information in analog format to digital information useful for further analyses and digital data sharing.

## 2 Materials and Methods

The Harmonized Soil Database of Ecuador (HESD) was developed by integrating information collected in previous projects: Generación de Geoinformación para la Gestión de territorio y valoración de tierras rurales de la Cuenca del Río Guayas, escala 1:25.000" (2007-2015) (CLIRSEN, 2015) by the Instituto

Espacial Ecuatoriano (IEE), and "Generación De Geoinformación para La Gestión Del Territorio A Nivel Nacional" (2009-2012) by the Ministerio de Agricultura y Ganadería within el Sistema Nacional de Información de Tierras Rurales e Infraestructura Tecnológica (MAGAP-SIGTIERRAS) (Tracasa-Nipsa,

2015). As a result, 13 542 soil profiles were described and registered, from which 5368 are from IEE and 8174 profiles from MAGAP-SIGTIERRAS (Figure 1).


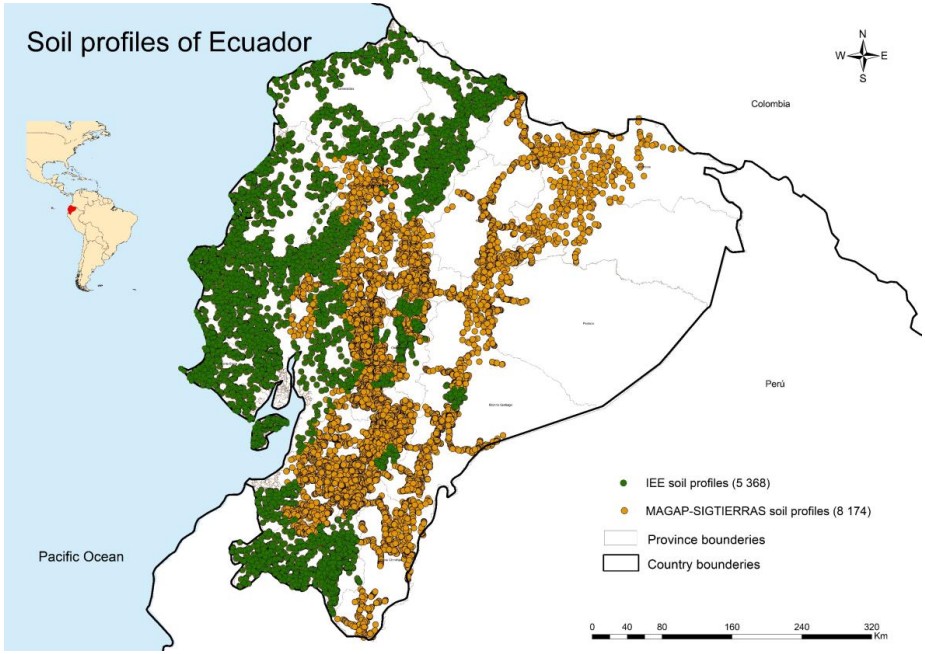

**Figure 1. Spatial distribution of soil profiles in Ecuador compiled in the HESD**

The original IEE data was available as a collection of portable document format (PDF) files, where each
PDF represented one soil profile containing morphological and analytical information. In contrast, soil morphological and analytical information from MAGAP-SIGTIERRAS was stored in different files in PDF format. We unified the information from IEE and MAGAP-SIGTIERRAS into one harmonized database (Figure 2). Given the size of the database, manual extraction of the original information was not feasible. Therefore, we developed an automated workflow using two programming languages (i.e., Python and R)
to optimize data extraction from the original datasets.



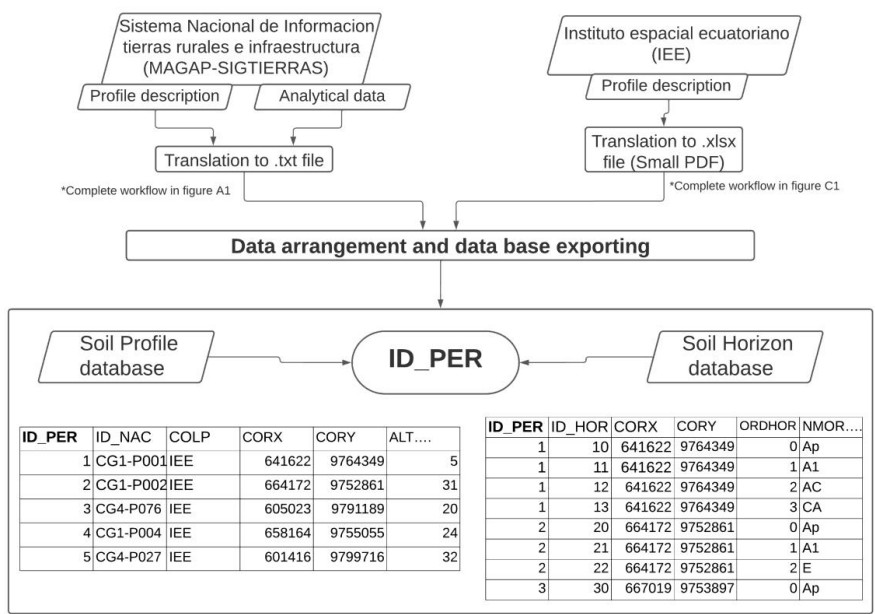

**Figure 2. Overview of the workflow for extracting data and structure database harmonized.**

### 2.1 Extracting Data from PDF files

Each available soil profile was divided into two groups depending on its original source (i.e., IEE or MAGAP-SIGTIERRAS). Specialized data handling libraries such as pandas (Wes McKinney 2011), openpyxl (Python Software Foundation, 2010), or pdf tools (Tracker Software Products, 2011) were used to automate this task. The first step to extract data was to convert the information from PDF format to a data format such as .xlsx or .txt. The data extracted contained categorical information about profile morphological description and tabular information with chemical and physical properties for each available soil horizon. The target information extracted for MAGAP-SIGTIERRAS, or IEE was organized using the Pandas Python Library and exported to HESD presented in this manuscript.

Data from MAGAP-SIGTIERRAS presented a homogeneous structure through, which simplified data extraction. The structure from the IEE information presented many irregularities that varied across the collection. Irregularities included: the number of fields and variables in the tables, table headers, and differences in categorical or descriptive fields. The heterogeneity of the structure in MAGAP-SIGTERRAS and IEE hindered the design of a homogeneous extraction methodology, therefore we applied two approaches as explained below.

### 2.1.1 MAGAP-SIGTIERRAS Approach



The homogeneous structure of MAGAP-SIGTIERRAS dataset allowed the development of a methodological approach based on regular expression queries. Each query sought a target variable or
information contained in the text.

First, all files from MAGAP-SIGTIERRAS were stored in a specific directory. Then, iteratively, each file was converted into a .txt file, preserving the format of the tables, using the R package 'pdftools' (Ooms, J., 2022). Once the files were converted, regular expressions were applied over the text to extract the key
variables, to perform this process own scripts were used, needing adaptation depending on the structure of the original database (Supplement A). The regular expression-based queries were imported in a data frame that held the information for a single file. Next, the resulting data frame was appended to a target data frame (i.e., final data frame) that contained all the processed information from all available files. Once all the files were processed, the final data frame was converted to a .csv file.

**2.1.2 IEE Approach**

Here, we aimed to convert the information stored in the pdf (text and tables) to a .xlsx format, where each sheet contained the text blocks or tables of the original pdf document. This format was use since this process was done with the free access program Smallpdf v 0.19.1 and it was the only option to extract the information. In this way, each sheet corresponded to the description of a group of morphological, chemical,
or physical properties of the soil.

Not always the conversion was successful, and many anomalies could be found on the table structures or sheet content. The inconsistencies in the conversion were due to the poor structure of the original data. Usually, the errors were related to merged rows, joint characters inside the variable descriptions, inconsistent labeling of the tables, or a different number of tables per file. Therefore, a Python 3.10.2 script
was generated to overcome these difficulties and successfully extract the data. The goal was to read the .xlsx files and transfer the information into another file whose tables were designed with the target structure of the HESD (Supplement D). To identify the errors, the scripts included an error handling system where a log .txt file was compiled containing information of the original file and tables that could not be converted. This procedure helped to identify problematic data files and track the evolution of the data extraction
process.

The rationale of the script was to generate a data frame for every sheet in an .xlsx file, where each sheet corresponds to a table with chemical or physical description. The target columns were identified for each table, and their information was passed to a dictionary that constructed the file data frame. After creating a data frame for each table, all the data frames were merged in a standard data frame for the .xlsx file; finally,
the file data frame was appended into a general data frame that contained the information for all the .xlsx files. Later the files were converted to format .csv to handle them in the next phase of correction and harmonization. Scripts and diagrams explaining the methodology used for each case can be found in the Supplements (B, D).



**2.2 Soil data correction and harmonization**

All the data obtained from the original sources went through a manual review process by an expert pedologist to minimize the data extraction errors and provide a curated harmonized dataset. Once the original databases were merged, the two subsets of the final database (profile information subset and horizon information subset) were manually revised a second time by the expert to detect any potential errors and inconsistencies. All fields in the database were checked using basic descriptive statistics, such as

minimum, maximum, average, and standard deviation values to verify the consistency of the data and the soil properties (e.g., pH range, CN ratio). In some fields it was necessary to make changes in the units of measurements in the harmonization tasks, either by standardizing the original datasets (i.e., IEE and MAGAP-SIGTIERRAS) or converting all units to the International Metric System. The variables "organic carbon" (CO), "organic matter" (MO), and "total nitrogen" (NTOT) were transformed to $g \cdot kg^{-1}$. The level

of precision in the expression of each variable was standardized (maximum of two decimals). Finally, some errors were found and corrected, such as duplicated information, missing data, errors in the information's agreement with the horizon, and formatting typos.

Special attention was paid to the quantitative information of the analytical variables, for which their frequency histograms were plotted to identify outliers or physical inconsistencies, such as excessively low

pHs (i.e., <3), extremely high Carbon/Nitrogen ratios (i.e., >35), or zero-value assignment in unrealized determinations. All inconsistencies that could not be resolved were reclassified as "without data".

**3. Overview soil dataset**

HESD contains information from 13 542 soil profiles with over 51 713 measured soil horizons, including 92 different edaphic variables. Over 4.7 million records that include numeric (e.g., clay content, organic

material, soil pH) and class (e.g., horizon designation, geology) soil properties represent the most complete data compilation for mainland Ecuador.

The structure of the database compilation is based on the Soil Organic Carbon Mapping Cookbook (FAO, 2018), and represents a complete soil data compilation for Ecuador, considering the effective soil depth (ESD). The ESD considers the solum, which includes surface and subsurface horizons with presence of

roots and biological activity (Soil Survey Staff, 1975) of the soil profile. Given the impossibility of designing a single structure for coupling the profile and the soil horizons information, the data was divided into two datasets linked by a unique identifier. Thus, the use of a relational database can easily be queried and augmented for future synthesis studies.

The common identifier linking these dataset tables is the ID_PER field, which records the unique name

assigned to the database. Both files (.csv) can easily be imported into statistical software such as R, after which they can be joined using the unique ID_PER. The first dataset contains information associated with the soil profile and its environmental characteristics (Table 1). It shows the variables in the profile dataset, with soil profile information (classification, humidity and temperature regime, rockiness, adequate depth)



and site-level data, containing the environmental information (forming factors): landscape attributes, land
cover type, slope.

The second dataset contains information associated with the soil profiles divided into horizons and
including qualitative and quantitative information. The dataset contains morphological information such as
designation or depth of horizon, presence or absence of roots, an abundance of rock fragments. In addition,
there are more than 30 variables related to soil physical properties (e.g., textural and bulk density) and
chemical properties (Table 2). We highlight that there is information regarding soil organic fraction, cation
exchange capacity, electrical conductivity and sodium exchange capacity, and soil properties (e.g., soil
drainage, soil tilth) relevant to evaluate soil health (USDA, 2022).

**4 Exploratory analyses of HESD**

We performed an exploratory analysis of some variables included in HESD as an example of the
characteristics of this database.  Soil variables behave differently when the soil depth increases, Fig. 3
shows examples of soil properties and depth relationships (SOC, soilPh-H2O (Ph), soil electrical
conductivity, clay, soil cation exchange capacity (CIC) and soil profile of effective depth (PRES)). For
example, SOC has higher values at the surface, and it gradually decreases as soil depth increases. In
contrast, pH ranges between 6 and 7 with an average of ~6.5 and this value is maintained as soil depth
increases. That said, we provide examples on how different soil properties vary as soil depth increases (Fig.
3).

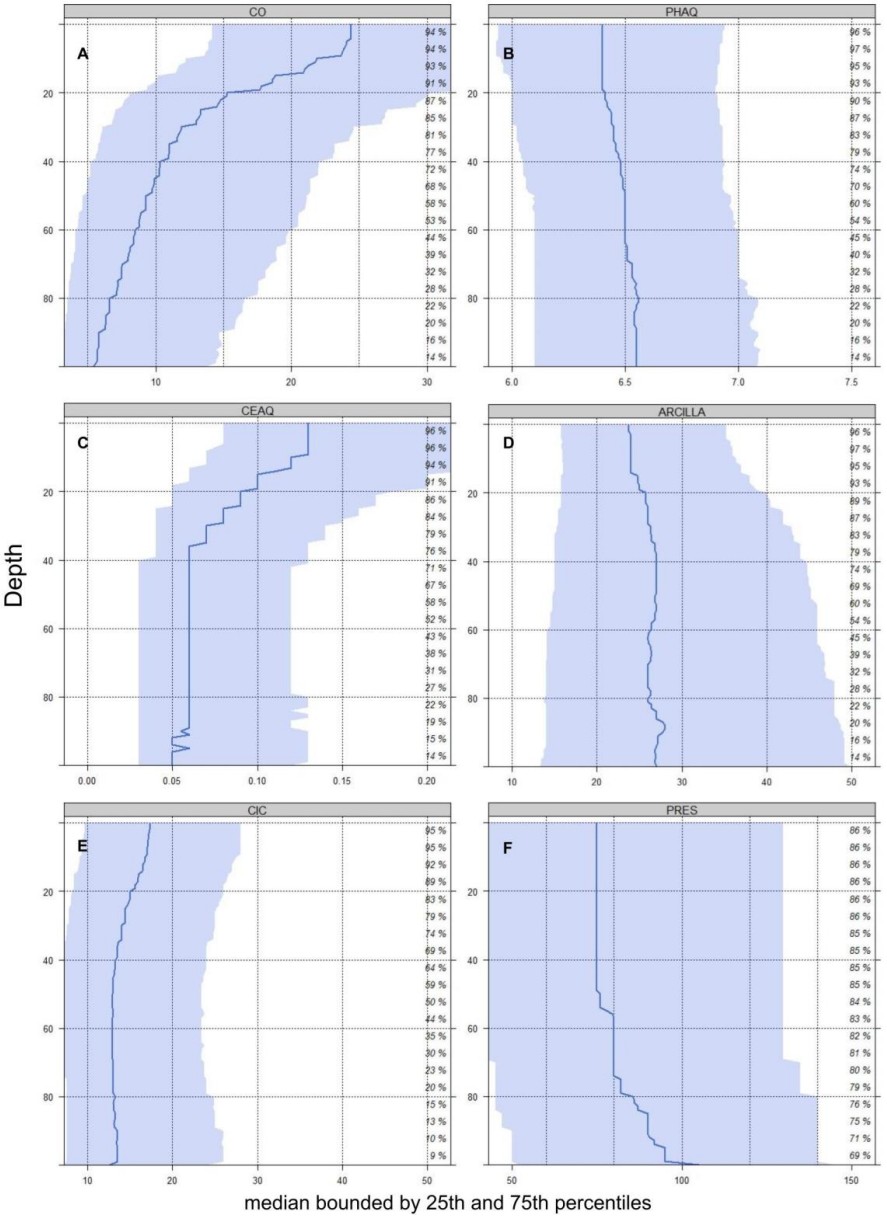

**Figure 3.** Variation of the concentration of soil variables with respect to its depth. **A.** Average profile of organic carbon (CO), **B**. Average profile of Ph H2O, **C**. Average profile of Electric conductivity in water (CEAQ), **D**. Average profile of electric conductivity in water total clay (ARCILLA), **E.** Average profile of cation exchange capacity (CIC), **F.** Average profile of effective depth (PRES). The blue area represents the range in which the properties oscillate.




Information in HESD could be used to evaluate how land use and management could affect soil properties
(Beillouin, et al., 2022). Table 3 shows a statistical analysis of different variables within two different

ecosystems: cropland and forest. Although HESD presents the most complete information at the national
level, we recognize that there are still information gaps. The two original projects from which the soil
information was extracted were focused on agricultural areas, for this reason it is assumed that HESD does
not fully represents all ecosystems across Ecuador. We highlight that there is bias in the data; for croplands,
HESD has 9675 points, and for forest, only 3694. With this in mind, the forest ecosystem presents a higher

average concerning SOC (CO, 27.9 g.kg).

**5 Spatial distribution and environmental representativeness of the database**

Two different analyses were made with HESD one focused on the representativeness of the data within the
different biogeographical sectors and a second focused on the probability of the spatial representativeness
at the national level. To do this, we used the Maximum Entropy approach (Maxent program; Phillips, et al.,

2020), which has been applied for assessing the spatial representativeness of environmental observatory
networks (Villarreal, et. al., 2019; Villareal et. al., 2018).

**5.1 Representativeness index by Ecuadorian Biogeographic Sectors**

The first analysis to test the representativeness was done considering the 15 biogeographic sectors of
Ecuador (Figure 4). We clarify that each biogeographic sector represents a group of plant communities that

share flora affinity in a genus and mainly at the species level, and thus define homogeneous environmental
units (Ministerio de Ambiente del Ecuador, 2013).

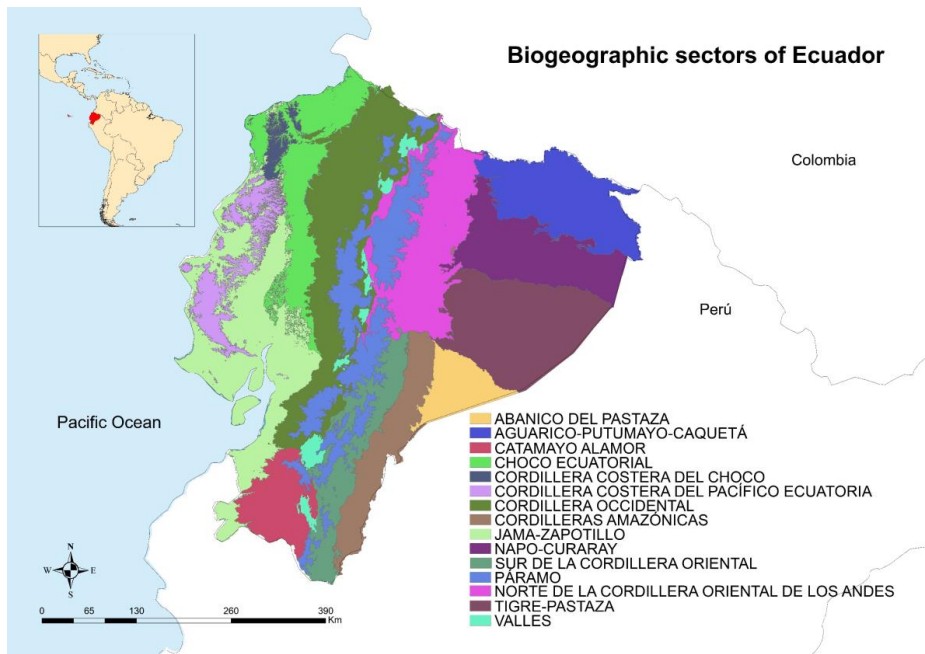

**Figure 4. Biogeographic sectors of Ecuador. Extracted from the "Sistema de clasificacion de Ecosistemas del Ecuador Continental "(Ministerio de Ambiente del Ecuador, 2013).**

We calculated the representativeness index for each sector based on the number of data points divided by the total coverage percentage of each biogeographic sector; where the higher the representativeness index, the better represented it is in the database (Pfeiffer et al., 2020). Table 4 shows the number of data compiled in this work, by region, province, biogeographic sector, and the representativeness index for each biogeographic sector.

The biogeographic sector with a higher representativeness index is Cordillera Occidental de los Andes with 24,7 %; followed by Jama-Zapotillo (16.7%), Norte de la cordillera Oriental de los Andes (11.4%), Sur de la Cordillera Oriental de los Andes (9.7%), and Paramo (7.6%) (Table 4). These areas are found mainly in the western part of Ecuador. The last four biogeographic sectors are grouped in what we call the Andes del Norte province in the Andes region. In Ecuador, this zone encompasses the Andes Mountain range that

extends from north to south (Clapperton 1993). In terms of SOC, these regions present the highest mean values (27,8g/kg).

The Andes, in the biogeographic sector of Paramo, has a SOC mean of 45 g/kg. This sector is distributed in a valley almost uninterrupted over the forest line of the eastern and western mountain ranges of the Andes (Hofstede et al. 1999) around 3 700 and 3 400 masl. This biogeographic sector occupies 23 452 km$^2$ (9.4

% of the national territory) (Table 4) and is probably the largest soil carbon reservoir in Ecuador. Despite the importance of Paramo as a large pool of SOC, it representativeness index is not as high as we expected



(109.8) probably because a large part of the area is within some of the national protected areas, zones that were not considered by the original projects.

Most of the data are concentrated in the southwest part of the country. In contrast, no soil data are available
for the eastern section of the country, mainly in the Amazonian region (31.4 of representativeness index), but the mean of carbon (17.7g/kg) in this region is higher than the Litoral region (3 579 observation, 15.5 g.kg SOC). This may be because it is known that the organic soil of the tropical forest is no deeper than 10 cm limiting carbon accumulation in soil (Hofstede, 1999). After all, the decomposition of the litter is so rapid that the plant material reaching the soil surface is, in most cases, oxidized before it could be
incorporated into the soil matrix.

**5.2 Spatial representativeness using the Maxent approach**

The second analysis carried out was performed using the Maxent approach (Yackulic et al., 2012). This analysis provides an estimate between 0 and 1 of probability of presence, and we interpret it as the probability of an area for being represented. This analysis allowed us to compare the spatial
representativeness of the HESD with the soil information currently available in WoSis (Betjes, et al., 2019), and we demonstrate how HESD contributes to filling the information gaps across Ecuador. Areas, where the values of soil information are minimized, at the national level improved with the HESD (Figure 5), this is very evident in the part of the coast and in the highlands.  As evidenced in Table 4, there are areas not yet fully represented with the data in HESD; this is the case in the eastern part of the country (Amazonia)
and in a part of the Esmeralda province (northwest), but it is evident a greater representativeness compared to the one that existed with the database of WoSis.



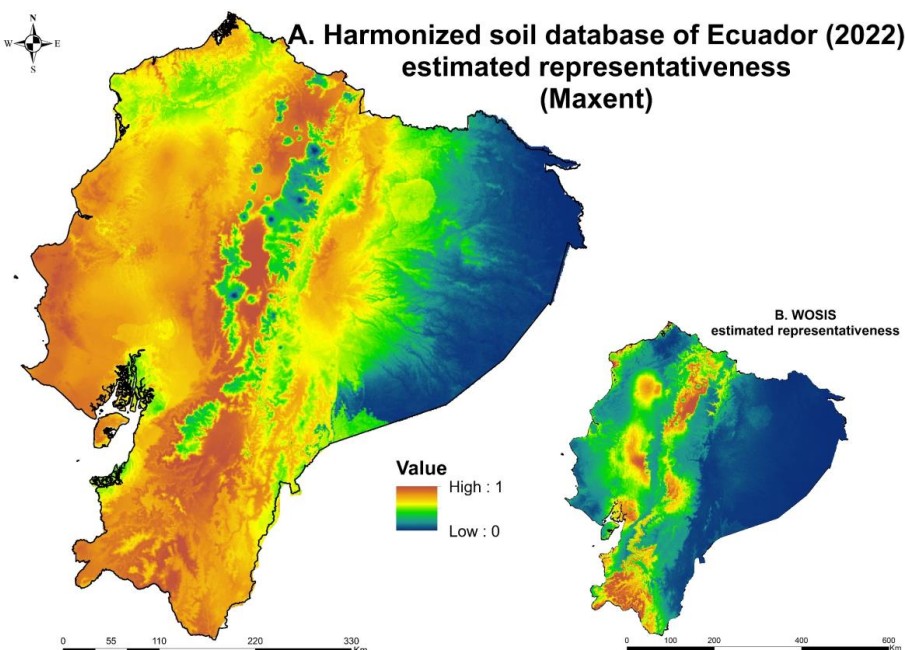

**Figure 5.** National representativeness of soil information using the HESD (a); and information available in WoSis (b).

The HESD shows a clustered distribution with some areas better represented than others due to the methodology used in the original projects that was biased (Table 4). We highlight that the original soil collection efforts (i.e., IEE and MAGAP-SIGTERRAS) were not focused on biogeographical sectors but rather focused on populated areas or areas designated for agriculture and did not consider protected areas. It is evident, through the two representativeness analyses that there are still areas that are not fully represented, such as the Choco Coastal Mountain Range sector (29.3%, coastal region) and all sectors in the Amazon region (Table 4 and Figure 5). We recommend that the next soil data raised at national level be added to HESD to keep it updated and gradually fill those gaps, and so represent a more certain reality.

**6 Data availability**

Data are available at https://doi.org/doi:10.6073/pasta/1560e803953c839e7aedef78ff7d3f6c Armas, et al., 2022), here are the two datasets (.csv files). which have a unique identifier (ID-PER) to link the profile information with the information of each horizon. Geographical coordinates are according to the UTM WGS 84.



## 8 Further Considerations


The HESD aims to increase the quantity, quality, and access to soil information across Ecuador. HESD facilitates the exchange and use of soil data collected within the context of collaborative efforts at a different scale (global, national, and local). As a result, HESD is a relational database composed of two independent datasets but linked by a unique identifier.

The proposed methodology demonstrates the possibility of rescue soil information previously stored in formats that are not easily accessible for data analysis. We propose a systematic method to help in the organization of national soil information and reduce errors when generating new data in the future (Yigini et al., 2018; Baritz et al., 2008). We substantially improved the publicly available spatial representation of soil information in Ecuador to support current soil information initiatives such as the WoSis (Batjes, et al.

2019), the Global/SoilMap.et project, and the FAO Global Soil Partnership to increase the access of soil information across the world. HESD includes information of more than 70 edaphic properties for Ecuadorian soils. It is evident that data gaps exist in certain areas and there is a need to incentivize for a future soil survey program to increase the sampling in underrepresented areas. HESD could support the generation of new soil-related knowledge to support food production challenges, threats to soil security and

soil health, climate change mitigation, and land degradation.

**Author contribution:** Daphne Armas, Mario Guevara and Cecilio Oyonarte work in the conceptualization and methodology of the paper, Fernando Bezares and Pilar Durante developed the code and scripts to extract the soil information, Rodrigo Vargas and Víctor Osorio worked in the writing – review & editing, , Wilmer Jiménez help with the original resources, Cecilio Oyonarte algo contributed with the funding acquisition,

Daphne Armas prepared the manuscript with contributions from all co-authors.

**Competing interests:** The authors declare that they have no conflict of interest."

Acknowledgments

This work has been carried out within the framework of the project "Desarrollo metodológico para la elaboración de un Sistema de Información de Suelos Global, y modelización de propiedades edáficas de

interés ambiental y agronómico.", financiado por la AACID (2017DEC003)/Junta de Andalucía.  Mario Guevara acknowledges support by the Programa de Apoyo a Proyectos de Investigación e Innovación Tecnológica (PAPIIT) under grant number IA204522. Rodrigo Vargas acknowledges support from NASA Carbon Monitoring System grant number (80NSSC21K0964).

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

       documents/card/en/c/I8895EN (last access: 16 July 2018).








TABLES

**Table 1**. HESD profile's variables names, codes, description and units

| Code | Property | Units | Description |
|---|---|---|---|
| ID_PER | Profile identifier | Unique | Unique profile identifier |
| ID_NAC | Profile identifier in the provenance collection | Unique | Profile id of the source project |
| COLP | Source project | | Name of the source project |
| CORX | Longitude coordinates | utm | Longitude UTM WGS84 projection |
| CORY | Latitude coordinates | utm | Latitude UTM WGS84 projection |
| ALT | Altitude | mts | meters above sea level |
| STSG | Classification Soil Subgroup | Nominal class | Soil Taxónomy[1] Soil Subgroup |
| STGG | Soil Grate group | Nominal class | Soil Taxonomy Soil Grate group |
| STOD | Soil Order | Nominal class | Soil Taxonomy Soil Order |
| RTS | Soil temperature regime | Nominal class | Soil Taxonomy soil temperature regime |
| RHS | Soil humidity regime | Nominal class | Soil Taxonomy soil humidity regime |
| PRES | Effective Depth | cm | Solum depth, according to field description |
| LITO | Litology | Nominal class | Lithological classes established on the geological map |
| GEOF | Geoform type | Nominal class | Landforms established on the geological map |
| PEND | Local pending | % | Slope of the sampling site |
| TUSO | Land use | Nominal class | Land use |
| TVEG | Type of vegetation | Nominal class | Field description using the model legend. coverage data |
| ROCS | Rock outcrops | % | Exposures of bedrock are described in terms of surface cover. The average value of the class established in GSD[2] |
| FRGG | Coarse surface fragments gravimetry | % | Surface coverage of rock fragments. Average value of the class established in GSD[2]. |
| TERO | Erosion type | Nominal class | Classification of erosion, by category established in GSD[2]. |
| GERO | Degree of erosion | Nominal class | Intensity of the erosion process, by category established in GSD[2] |
| DREN | Drainage conditions | Nominal class | Drainage conditions by category established in GSD[2]. |
| FEMU | Soil sample date | dd/mm/yyyy | Profile sampling date |

[1] USDA soil taxonomy (ST) developed by United States Department of Agriculture and the National Cooperative Soil Survey

[2] Guidelines for soil description Fourth edition. FOOD AND AGRICULTURE ORGANIZATION OF THE UNITED NATIONS

(FAO). Rome, 2006

**Table 2.** HESD Horizons coding conventions and soils property names and their description, units of measurement

| Code | Property | Units | Description |
|---|---|---|---|
| ID_PER | Profile identifier | Unique | Unique profile identifier |
| ID_HOR | Horizon identifier | Unique | Unique numeric identifier of the horizon |
| CORX | Longitud coordenates | utm | Longitud UTM WGS84 projection |
| CORY | Latitud coordenates | utm | Longitud UTM WGS84 projection |
| *Morphological properties* | | | |
| ORDHOR | Horizon number | - | Horizon position in profile sequence |
| HMOR | Morphological horizon | - | Completed morphological soil horizon designation, , according to GSD[2]. |
| MSHOR | Master horizon | - | Designation master horizons, according to GSD[2]. |
| SUBHOR | Subordinate characteristic | - | Subordinate characteristics within master horizons, according to GSD[2]. |
| DISHOR | Discontinuities | - | Numerals used as prefixes to indicate discontinuities |
| LIMSUP | Upper boundary of horizon | cm | |
| LIMINF | Lower boundary of horizon | cm | |
| ROOTS | Roots | presence / absence | Presence of roots in the field description |
| FR_CL | Rock fragments/qualitative | abundance range | Rock fragments (> 2 mm). The abundance class limits, by volumen, correspond with GSD[2]. |
| FR_QT | Rock fragments/quantitative | % | Abundance large rock, by volume, expressed as the mean of the intervals of GSD[2]. |
| *Physical properties* | | | |
| ARENA | Sand total | % | Proportion of sand-size particles, by weight, USDA[3] textural classes. Bouyoucos method |
| LIMO | Silt total | % | Proportion of silt-size particles, by weight, USDA textural classes. Bouyoucos method |
| ARCILLA | Clay total | % | Proportion of clay-size particles, by weight, USDA textural classes. Bouyoucos method |
| DA | Bulk density | $g.cm^{-3}$ | Bulk density of the fine-earth fraction, air dried |
| *General chemical properties* | | | |
| PHAQ | pH $H_2O$ | - | Measure of the acidity in a soil/water solution (1:2.5) |
| ACINT | Exchange acidity | $cmol.kg^{-1}$ | Volumetric |
| ALINT | Exchange aluminum | $cmol.kg^{-1}$ | Volumetric |
| NAM | Amonical nitrogen | $mg.kg^{-1}$ | Measured according to the Olsen method modified (pH 8.5) |
| PDIS | Available phosphorus | $mg.kg^{-1}$ | Measured according to the Olsen method modified (pH 8.5) |



| KDIS | Available potassium | cmol.kg⁻¹ | Measured according to the Olsen method modified (pH 8.5) |
|---|---|---|---|
| CADIS | Available calcium | $cmol.kg^{-1}$ | Measured according to the Olsen method modified (pH 8.5) |
| MGDIS | Available Magnesium | $cmol.kg^{-1}$ | Measured according to the Olsen method modified (pH 8.5) |
| CEAQ | Electric conductivity in water | $dS.m^{-1}$ | Electric conductivity of a 1:2.5 soil–water extract |
| MO | Organic matter | $g.kg^{-1}$ | Gravimetric content of organic matter. Calculated multiplying by factor 1.72 the OC content |
| CO | Organic carbon | $g.kg^{-1}$ | Gravimetric content of organic carbón.Measured using wet-oxidation method (Walley-Black) |
| NTOT | Total nitrogen | $g.kg^{-1}$ | The sum of total Kjeldahl nitrogen |
| CN | Carbon/Nitrogen relation | - | |

### Soil cation exchange complex

| CIC | Cation exchange capacity | $cmol(c).kg^{-1}$ | Capacity to hold exchangeable cations, estimated by ammonium acetate buffering to pH:7 |
|---|---|---|---|
| NACC | Exchangeable sodium | $cmol.kg^{-1}$ | Sodium hold in the exchange complex, estimated by ammonium acetate buffering to pH:7 |
| KCC | Exchangeable potassium | $cmol.kg^{-1}$ | Potassium hold in the exchange complex, estimated by ammonium acetate buffering to pH:7 |
| CACC | Exchangeable calcium | $cmol.kg^{-1}$ | Calcium hold in the exchange complex, estimated by ammonium acetate buffering to pH:7 |
| MGCC | Exchangeable magnesium | $cmol.kg^{-1}$ | Magnesium hold in the exchange complex, estimated by ammonium acetate buffering to pH:7 |
| SBCC | sum of bases in exchange complex | $cmol.kg^{-1}$ | Sum of cations determined in the exchange complex |
| SATCC | saturation of exchange complex | % | Percentage of exchange complex occupied by bases |

### Chemical properties of soil solution (Salinity)

| pHSS | pH in soil solution | - | Measure of the acidity in soil solution extracted by the saturated paste method (SPM) |
|---|---|---|---|
| CESS | Electric conductivity in soil solution | $dS.m^{-1}$ | Electric conductivity in soil solution (SPM) |
| NASS | Sodium in soil solution | $cmol.kg^{-1}$ | Sodium in soil solution (SPM) |
| KSS | Potassium in soil solution | $cmol.kg^{-1}$ | Potassium in soil solution (SPM) |
| CASS | Calcium in soil solution | $cmol.kg^{-1}$ | Calcium in soil solution (SPM) |
| MGSS | Magnesium in soil solution | $cmol.kg^{-1}$ | Magnesiun in soil solution (SPM) |
| SBSS | Sum of bases in soil solution | $cmol.kg^{-1}$ | Sum of cations determined in soil solution (SPM) |
| CARSS | $CO_3^=$ anion in soil solution | $cmol.kg^{-1}$ | Carbonate anion in soil solution (SPM) |
| SULSS | $SO_4^=$ anion in soil solution | $cmol.kg^{-1}$ | Sulfate anion in soil solution (SPM) |
| CLSS | $Cl^-$ anion in soil solution | $cmol.kg^{-1}$ | Chloride in soil solution (SPM) |



| PSI | Exchangeable sodium percentage | % | Extent to which the exchange complex of a soil is occupied by sodium |
| RAS | Sodium adsorption rate | - | Sodium adsorption rate (SAR), calculated from the concentrations of $Na^+$, $Ca^{2+}$ and $Mg^{2+}$ in soil solution (SPM) |

[2] Guidelines for soil description Fourth edition. FOOD AND AGRICULTURE ORGANIZATION OF THE UNITED NATIONS
(FAO). Rome, 2006

[3] The USDA system classifies soils into 12 soil texture classes.










**Table 3**. Statistical analysis of key variables in HESD. Farming 9675 – Forest 3694 data points.

| Variable | Mean | SD | CV | Max | Min |
|---:|---|---|---|---|---|
| CO | 25.65 | 25.28 | 0.98 | 277.03 | 0.05 |
| Cropland | 24.90 | 22.92 | 0.92 | 277.03 | 0.05 |
| Forest | 27.92 | 31.26 | 1.11 | 264.61 | 0.10 |
| PhAQ | 6.48 | 0.80 | 0,12 | 10.33 | 1.00 |
| Cropland | 6.45 | 0.76 | 0.11 | 9.90 | 1.00 |
| Forest | 6.54 | 0.90 | 0.14 | 10.33 | 1.00 |
| CEAQ | 0.29 | 0.51 | 3.20 | 225.00 | 0.01 |
| Cropland | 0.22 | 0.47 | 3.04 | 225.00 | 0.01 |
| Forest | 0.49 | 0.63 | 3.48 | 114.30 | 0.01 |
| ARENA | 40.91 | 18.18 | 0.44 | 97.00 | 0.28 |
| Cropland | 40.50 | 18.12 | 0.44 | 97.00 | 0.28 |
| Forest | 42.03 | 18.36 | 0.44 | 96.00 | 0.28 |
| ARCILLA | 29.19 | 17.58 | 0.59 | 96.00 | 0.36 |
| Cropland | 29.05 | 17.60 | 0.60 | 96.00 | 0.36 |
| Forest | 29.57 | 17.45 | 0.56 | 94.46 | 1.00 |
| CIC | 19.05 | 12.09 | 0.71 | 100.8 | 0.30 |
| Cropland | 18.63 | 11.81 | 0.69 | 101.8 | 0.40 |
| Forest | 20.20 | 12.90 | 0.77 | 98.86 | 0.30 |
| PRES | 85.08 | 48.54 | 0.56 | 220.00 | 0.05 |
| Cropland | 89.42 | 48.06 | 0.53 | 220.00 | 0.05 |
| Forest | 72.47 | 48.33 | 0.64 | 185.00 | 0.36 |

**CO** = Carbon, **PHAQ** = pH H2O, **CEAQ** = Electric conductivity in water, **ARENA** = Sand total,

**ARCILLA** = Clay total, **CIC** = Cation exchange capacity, **PRES** = Effective Dept





559  **Table 4.** Distribution of SOC data points per ecosystem sector (vegetation formation) according to

560  Ministerio del Ambiente del Ecuador) (2013).

561

| Region | Province | Sector | Data points | Data points (%) | Country area (km$^2$) | Country area (%) | Density (data/km$^2$) | Representativeness index (data per % area) | |
|---|---|---|---|---|---|---|---|---|---|
| Litoral | Choco | Choco Ecuatorial | 811 | 6,0 | 19 205 | 7,7 | 0,042 | 105,4 | |
| | | Cordillera Costera del Choco | 27 | 0,2 | 2 304 | 0,9 | 0,012 | 29,3 | 97,4 |
| | Pacifico Ecuatorial | Jama-Zapotillo | 2 255 | 16,7 | 35 252 | 14,1 | 0,064 | 159,7 | |
| | | Cordillera Costera Pacifico Ecuatorial | 486 | 3,6 | 9 341 | 3,7 | 0,050 | 129,9 | 137,1 |
| | | | 3 579 | | | | 0.054 | | 135,6 |
| Andes | Andes del Norte | Norte Cordillera Oriental de los Andes | 1 538 | 11,4 | 22 498 | 9,0 | 0,068 | 170,7 | |
| | | Sur Cordillera Oriental de los Andes | 1 314 | 9,7 | 12 877 | 5,2 | 0,102 | 254,8 | |
| | | Valles | 710 | 5,2 | 3 500 | 1,4 | 0,203 | 506,4 | |
| | | Páramo | 1 031 | 7,6 | 23 452 | 9,4 | 0,044 | 109,8 | |
| | | Cordillera Occidental de los Andes | 3 342 | 24,7 | 30 053 | 12,0 | 0,111 | 277,6 | |
| | | Catamayo-Alamor | 997 | 7,4 | 9 267 | 3,7 | 0,108 | 268,6 | |
| | | | 8 932 | | | | 0.088 | | 219,5 |
| Amazonía | Amazonía Noroccidental | Aguarico-Putumayo-Caqueta | 201 | 1,5 | 19 019 | 7,6 | 0.011 | 26,4 | |
| | | Napo-Curaray | 243 | 1,8 | 18 183 | 7,3 | 0.013 | 33,4 | |
| | | Tigre-Pastaza | 15 | 0,1 | 24 781 | 9,9 | 0.0006 | 1,5 | |
| | | Abanico del Pastaza | 47 | 0,3 | 7 262 | 2,9 | 0.006 | 16,2 | |
| | | Cordilleras Amazónicas | 525 | 3,9 | 12 659 | 5,1 | 0.041 | 103,5 | |
| | | | 1 031 | | | | 0.013 | | 31,4 |

562