# Peer review of "Harmonized soil database of Ecuador (HESD): data from 2009 to 2015"

_Earth System Science Data, 2022_

## Referee Comment (RC1)

*[handwritten notes: 1) units
2) long sentences should be simplified (eg. L 41-45)
3) more descriptive labels in tables & figure captions
4) Too much detail in data cleaning/scripts. Add these as sup. mat. and leave out some of section 2.]*

| | | | 1 031 | | | | 0.013 | | 31,4 |

562

[referee-annotated manuscript omitted]

---

## Author Response (AR1)

REFERRE 1

Comment Referee 1: Units
Author's response: Added units in figures and tables. In Figure 3 the unit cm was added to the depth, in table 3 units were added to all the variables

Comment Referee 1: Long sentences should be simplified (eg. L 41-45)
Author's response: Very long sentences have been simplified: "Furthermore, access to spatially explicit, consistent, and reliable soil data is essential for digital soil mapping and for evaluating the status of soil resources with increased resolution to respond and assess global issues" (see lines 41-43)

Comment Referee 1: More descriptive labels in table and figure
Author's response: Some labels were kept with abbreviations to maintain concordance with the complete tables, this is the case in figure 1

Comment Referee 1: Too much detail in data cleaning/scripts. Add this as sup. Material and leave out some of section 2
Author's response: The explanation of the scripts has been simplified. The explanation of the procedure has been simplified from line 172 to 173.

Comment Referee 1: Figure 2.  These tables are not meaningful, perhaps translate to human readable column headings
Author's response:  We believe that the table presentation is suitable to make evident the field (ID_PER) with which the two tables can be joined. The description has been modified to emphasize this aspect that we consider important (see lines 135-138)

Comment Referee 1: Exchange section 6 & 7
Author's response: Changed the following section to 7.

REFERRE 2

Comment Referee 2: Line 21: correct "how make it" for "how to make it".
Author's response: Corresponding correction was made

Comment Referee 2: Line 24: delete space in "13 542" and throughout in this kind of numbers.
Author's response: Corresponding correction was made

Comment Referee 2: Line 42: it should be "for digital soil mapping and for evaluating".
Author's response: Corresponding correction was made

Comment Referee 2: Line 59: it should be "and even less".
Author's response: Corresponding correction was made

Comment Referee 2: Line 64: delete "a" in "into a standard".
Author's response: Corresponding correction was made

Comment Referee 2: Line 71: I suggest rewriting this sentence as "Interoperability is defined as the collective effort of sharing information that can be used to produce and apply newly gained knowledge, and this is achieved by removing conceptual, technological, organizational, and cultural barriers".

Author's response: Suggestion is accepted, and the text has been modified "Interoperability is defined as the collective effort of sharing information that can be used to produce and apply newly gained knowledge, and this is achieved by removing conceptual, technological, organizational, and cultural barriers". (See lines 72-74)

Comment Referee 2: Line 78: it should be "and to evaluate".
Author's response: Corresponding correction was made

Comment Referee 2: Line 101: I suggest rewriting this sentence as "In this way, we developed a new soil database with the purpose of constituting a national soil information system following international standards for archiving and sharing soil data."
Author's response: Suggestion is accepted, and the text has been modified (see lines 104-106)

Comment Referee 2: Line 108: I suggest giving a translation of the projects' names and of the Ministry. This will give the reader a better idea of what the projects are about.
Author's response: Suggestion is accepted we added the names in ingles, "Generation of Geoinformation for land management and rural land valuation in the Guayas River Basin, scale 1:25,000" (2007-2015) "and "Generation Of Geoinformation For The Management Of The Territory At National Level" (2009-2012)" (See lines 112-116)

Comment Referee 2: Figure 1: The insert with the American map would look better with an outline, same as in Figure 4. Also, province names are too small to read. I suggest erasing them anyways.
Author's response: Figure was modified, province names were erased, the outline were added same as in Figure 4.

Comment Referee 2: Figure 2: The figure description should say "harmonized database structure". Also, the figure is not easy to understand. In the bottom half of the figure, I suggest adding more informative names for column headers, and delete the example data. This figure should give more an idea of the structure and the meaning of variables in the database. Avoid using abbreviated names for each variable unless you explain the meaning in the figure description. More detailed information on the actual configuration of the dataframe file should be given in supplement material.
Author's response: The description has been modified to incorporate the referee's suggestion: Harmonized database, structure, and overview of the workflow for extracting data.
It is preferred to keep the figure since it coincides with the current configuration of the dataframe file, but we have added the description of each abbreviation and emphasized the ID_PER field for better understanding. (See lines 134-138)

Comment Referee 2: Line 132: delete "Wes" since it's his first name and correct this accordingly in the References section.
Author's response: The corresponding modification was made

Comment Referee 2: Line 138: delete "presented in this manuscript".
Author's response: The corresponding modification was made

Comment Referee 2: Line 139: delete "through"
Author's response: The corresponding modification was made

Comment Referee 2: Line 155: Did you mean "we used our own scripts" or something else in the lines of "we used scripts written for this purpose"? Either way, modify this sentence accordingly.

Author's response: The corresponding modification was made. (See lines 164-166)

Comment Referee 2: Line 162: I would change it for "Our only option to extract the information with this format was the free access program Smallpdf v 0.19.1".

Author's response: The corresponding modification was made. (See lines 173-174)

Comment Referee 2: Line 202: Change this either for "Soil dataset overview" or "Overview of soil dataset"
Author's response: Switched to Soil dataset overview

Comment Referee 2: Line 204: Change this sentence for "With over 4.7 million records that include numeric (e.g., clay content, organic material, soil pH) and class soil properties (e.g., horizon designation, geology), HESD represents the most complete data compilation for mainland Ecuador".
Author's response: Suggestion is accepted: " With over 4.7 million records that include numeric (e.g., clay content, organic material, soil pH) and class soil properties (e.g., horizon designation, geology), HESD represents the most complete data compilation for mainland Ecuador". (See lines 208-210)

Comment Referee 2: Line 223: it should be "and abundance".
Author's response: The corresponding modification was made

Comment Referee 2: Line 227: it should be "relevant for the evaluation of soil health"
Author's response: The corresponding modification was made

Comment Referee 2: Figure 3: In the description of the figure, what do you mean by "average profile"? Do you mean "profile average"? Also, you should provide the units in the description and also in the figure itself. It is not clear what the percentages in the right-y-axis mean. Font size in general in this figure is too small to read.
Author's response: The corresponding modification was made. Units were added

Comment Referee 2: Line 243: Try to make more connections between these lines in the paragraph. It seems very disconnected as it is now. I suggest: "Information in HESD could be used to evaluate how land use and management could affect soil properties (Beillouin et al., 2022). As an example, Table 3 shows a statistical analysis of different variables within two different ecosystems: cropland and forest. Although HESD presents the most complete 245 information at the national level, we recognize that there are still information gaps. One of the reasons behind this is that the two original projects from which the soil information was extracted were focused on agricultural areas, for this reason it is assumed that HESD does not fully represent all ecosystems across Ecuador. Further, we emphasize that there is bias in the data since croplands have 9675 points and forests, only 3694. With this in mind, the forest ecosystem presents higher average SOC (27.9 g. kg-1)."
Author's response: Suggestion accepted; the change was made from line 247 to 255

Comment Referee 2: Line 252: add a "," after HESD.
Author's response: The corresponding modification was made

Comment Referee 2: Line 260: it should be "at a genus".
Author's response: The corresponding modification was made

Comment Referee 2: Figure 4: in the legend it should be "Cordillera Costera Pacífico Ecuatorial"
Author's response: The corresponding modification was made in figure 4

Comment Referee 2: Line 267: it should be "data points compiled".
Author's response: Suggestion accepted

Comment Referee 2: Line 270: it should be "with the highest".
Author's response: Suggestion accepted

Comment Referee 2: Line 271: it should be "24.7 %".
Author's response: Suggestion accepted

Comment Referee 2: Line 275: it should be "27.8 g/kg".
Author's response: Suggestion accepted

Comment Referee 2: Line 281: it should be "its representativeness"
Author's response: Suggestion accepted

Comment Referee 2: Line 287: Did you mean "organic soil layer" instead of just "organic soil"?
Author's response: The corresponding modification was made

Comment Referee 2: Figure 5: Explain in the figure description what is the meaning of the index.
Author's response: The corresponding modification was made "Figure 5. National representativeness (an estimate between 0 and 1 of probability of presence) of soil information using the HESD (a); and information available in WoSIS (b)."

Comment Referee 2: Line 314: I suggest changing "and so represent a more certain reality" with "to better represent the entire geographical range of Ecuador" or something along these lines.
Author's response: Suggestion accepted "to better represent the entire geographical range of Ecuador" (see line 320)

Comment Referee 2: Line 316: correct the citation format, it should be "(Armas et al., 2022)".
Author's response: The corresponding modification was made

Comment Referee 2: Line 323: can you expand on the global scale aspect?
Author's response: Globally HESD has the structure to be considered for use in different international projects including the Global organic carbon Map (GSOCmap) a project of FAO and the Global Soil Partnership (GSP) and the GlobalsoilMap.net Project. (See lines 327-331)

Comment Referee 2: Line 325: it should be "possibility to rescue".
Author's response: Suggestion accepted

Comment Referee 2: Line 334: is suggest changing "to support" with "which could help to assess".
Author's response: Suggestion accepted

Comment Referee 2: Line 336: it should be "Oyonarte worked".
Author's response: Suggestion accepted

Comment Referee 2: Line 339: it should be "Jiménez helped".
Author's response: Suggestion accepted

Comment Referee 2: Line 341: delete final ".
Author's response: The corresponding modification was made

Comment Referee 2: Line 345: "financiado por la" is in spanish, please translate this.
Author's response:  The corresponding modification was made "financed by"

Comment Referee 2: Line 476: Wes is his first name, this should be McKinney W.
Author's response: The corresponding modification was made, deleted Wes

Comment Referee 2: Table 1: the units in Altitude should be masl. Also, correct "Local pending" for "Local slope".
Author's response: The corresponding modification was made

Comment Referee 2: Line 515: Use a proper citation style here.
Author's response: The corresponding modification was made

Comment Referee 2: Table 2: the description of the table should be "HESD Horizons coding conventions and soils property names, units of measurement and their description". Also, under Organic carbon, correct the variable description with "organic carbon. Measured". Under Exchangeable sodium, potassium, calcium, and magnesium correct "hold" for its past participle form "held".
Author's response: The corresponding modification was made in table 2, the description of the table was changed, and the suggested corrections were made.

Comment Referee 2: Table 3: Correct the meaning of the initials in "CO = organic carbon" and in "PRES= Effective Depth".
Author's response: The corresponding modification was made

Comment Referee 2: Table 4: In the table description, delete ")" before (2013). Also, the names of the sectors of a same province are difficult to tell apart. It is not clear where one name ends and another one begins. Maybe add another horizontal line between the names or try to make them as distinct as possible. Also, when you give values for a region in its totality, please specify that on the same row by adding the word "Total", for example.
Author's response: The corresponding modification was made in the table, Horizontal lines between the names were added. The Word "Total" was added.

REFERRE 3
Comment Referee 3: Line 68: Is "soil organic carbon (SOC)" same with "organic carbon(CO)"? Please explain clearly in the manuscript or use uniformly.
Author's response: We use "SOC" when talking about soil organic carbon and "CO" when talking about the variable within the database, which abbreviation is given in the official language of the HESD.

Comment Referee 3: Line 162: "This format was use since this process was done with the free access program"
Author's response: The corresponding modification was made in lines 173-174

Comment Referee 3: Line 245: Are there only two ecosystem types (cropland and forest) in Ecuador? What is their proportions?

Author's response: There are more than 20 ecosystems, as an example we chose the two most nationally representative ones

| Ecosystem | Área (ha) | % |
|-----------|-----------|---|
| Agrícola | 6,951,200 | 25,68 |
| Forestal | 12,093,300 | 44,68 |

Comment Referee 3: Figure 1: The left map is small, and the location of Ecuador is not clear, which could be marked with an arrow.
Author's response: Ecuador is in red color in the left map, added lines to make it more evident where the country is on the map.

Comment Referee 3: Figure 3: The unit of Depth should be given.
Author's response: Units were added in Figure 3

Comment Referee 3: Table 1, Table 2: The common units of Longitude and Latitude are.
Author's response: The coordinates are expressed in UTM units and are inherited from the original projects.

Comment Referee 3: Table 2: "Longitud" should be Longitude, "Latitud" should be Latitude.
Author's response: The corresponding modification was made

Comment Referee 3: Are sand, silt and clay separated with the diameter of soil particles? If so, the diameter ranges should be introduced in the manuscript.
Author's response: Size limit of soil fractions, following the classification proposed by the USDA
Sand: from 2 mm to 0.05 mm.
Silt: 0.05 mm to 0.002 mm.
Clay: particles smaller than 0.002 mm.
In table 2 we indicate the textural classification system used.

Comment Referee 3: What is "Amonical nitrogen"? More introduction in Description.
Author's response: Is a measure of the amount of ammonia (inorganic compound having the chemical formula $NH_3$) in a soil sample. In this case according to Olsen's method.
We have introduced the clarification in table2

Comment Referee 3: In "Calculated multiplying by factor 1.72 the OC content", OC or CO? Because Organic carbon is abbreviated into CO in the manuscript.
Author's response: The corresponding modification was made. We use the acronym "CO" since the names of the variables in the table are in Spanish "carbono orgánico" (organic carbon).

Comment Referee 3: Table 3: The units of the variables should be shown.
Author's response: Units were added in table 3

Comment Referee 3: Are the values in the rows of the variables the total of all data points?
Author's response: Yes,of the data points that had this information

Comment Referee 3: In "Farming 9675 – Forest 3694 data points", the symbol of " – " is confused. "Effective Dept" should be "Effective Depth".
Author's response: The corresponding modification was made. The spelling error is accepted and corrected (see line 562)

Comment Referee 3: Table 4: Please explain clearly the meaning of number before and after comma.

Author's response: The coma was a typo, and it was changed with a point to represent decimal units.

Comment Referee 3: The values could not correspond clearly with Sector.

Author's response: a horizontal line was added in table 4 to make this table easier to read.